

# Towards ubiquitous requirements engineering through recommendations based on context histories

Robson Lima[1], Alexsandro S. Filippetto[1], Wesllei Heckler[1], Jorge L.V. Barbosa[1] and Valderi R.Q. Leithardt[2,3]

[1] Applied Computing Graduate Program (PPGCA), University of Vale do Rio dos Sinos (UNISINOS), São Leopoldo, RS, Brazil
[2] VALORIZA–Research Centre for Endogenous Resource Valorization, Polytechnic Institute of Portalegre, Portalegre, Portugal
[3] COPELABS, University Lusófona–ULHT, Lisboa, Portugal

## ABSTRACT

The growing technological advance is causing constant business changes. The continual uncertainties in project management make requirements engineering essential to ensure the success of projects. The usual exponential increase of stakeholders throughout the project suggests the application of intelligent tools to assist requirements engineers. Therefore, this article proposes Nhatos, a computational model for ubiquitous requirements management that analyses context histories of projects to recommend reusable requirements. The scientific contribution of this study is the use of the similarity analysis of projects through their context histories to generate the requirement recommendations. The implementation of a prototype allowed to evaluate the proposal through a case study based on real scenarios from the industry. One hundred fifty-three software projects from a large bank institution generated context histories used in the recommendations. The experiment demonstrated that the model achieved more than 70% stakeholder acceptance of the recommendations.

## INTRODUCTION

In recent years, the continuous and growing use of new technologies results in a Digital Transformation, bringing disruptive changes across domains (*Nadkarni & Prügl, 2021*). The techniques considered crucial to eliciting requirements do not hold up, given the paradigm shifts that have occurred. *Villela, Groen & Doerr (2019)* argued that Requirements Engineering (RE) involves various dimensions and thus ubiquitous RE allows an adequate approach for handling the complexity involved.

The software has become present in the vast majority of businesses, with companies that lack some level of automation being rare. Enterprises need to deal with increasingly diverse, complex, and interconnected systems, while the demand for rapid innovations requires ever-shorter feedback loops. The spread of software in business-to-consumer

Corresponding author
Wesllei Heckler,
weslleiheckler@edu.unisinos.br

and business-to-business environments makes it difficult to engage the growing number of stakeholders. Traditional requirements elicitation techniques, such as interviews or focus groups, present problems of scalability and limitation when they need to occur continuously involving the growing number of stakeholders (*Villela et al., 2018*).

RE stands out as one of the most critical areas for software project results. Factors such as goal setting, project planning, involvement, and identification of user needs are key to project success (*Hastie & Wojewoda, 2015*). In the meantime, incorrect application of RE is a primary reason for project failures, increasing development time, and cost (*Dick, Hull & Jackson, 2017*; *Project Management Institute, 2017b*; *Bozyiğit, Aktaş & Kılınç, 2021*). When proper requirements management is applied, the chances of project success increase. Studies indicate that 30% of project success factors are related to RE processes (*Hastie & Wojewoda, 2015*). Apart from that, reusing requirements can help in the execution of projects, reducing the time for analysis of requirements and identifying reusable code and artifacts, such in case of software development (*Irshad, Petersen & Poulding, 2018*).

One option for addressing the issues faced by requirements engineers is requirements reuse. Software Engineering Recommendation Systems (SERSs) help teams select information and make decisions when they are inexperienced or unable to consider all available data. However, setting context is a challenge for recommendation systems (*Robillard et al., 2014*).

The use of ubiquitous computing (*Lopes et al., 2014*) is an alternative for assisting requirements engineers in their activities. The classical works of *Weiser (1999)*, *Satyanarayanan (2001)*, and *Dey, Abowd & Salber (2001)* defined the ubiquitous computing and context-aware computing. Since then, these concepts have been applied in different knowledge areas such in health (*Vianna & Barbosa, 2014*; *Vianna & Barbosa, 2019*; *Dias et al., 2020*; *Petry et al., 2020*; *Bavaresco et al., 2020*), well-being (*Vianna, Barbosa & Pittoli, 2017a*), competence management (*Rosa et al., 2015*), learning (*Barbosa et al., 2011*; *Wagner, Barbosa & Barbosa, 2014*; *Barbosa et al., 2014*; *Larentis et al., 2020*), commerce (*Barbosa et al., 2016*), accessibility (*Tavares et al., 2016*; *Barbosa et al., 2018*), Smart Cities (*Rolim et al., 2016*; *Orrego & Barbosa, 2019*; *Matos et al., 2021*), and agriculture (*De Souza et al., 2019*; *Bhanu, Reddy & Hanumanthappa, 2019*; *Helfer et al., 2020*). The application of ubiquitous computing in project management coined the term Ubiquitous Project Management (*Filippetto et al., 2020*).

The ubiquitous computing is aware of contexts and allows to use this information to introduce context awareness in the computational systems. Based on contexts, the systems adapt the execution according to the strategic information obtained in the runtime (*Abech et al., 2016*). Recently, the use of context-aware computing to support the development and maintenance of software emerged as a strategic research theme (*D'Avila, Barbosa & De Oliveira, 2020*; *D'Avila, De Oliveira & Barbosa, 2020*). In addition, in disruptive applications the ubiquitous computing has been considered an alternative to develop hygge software (*Vianna, Barbosa & Pittoli, 2017a*). As a recent evolution, ubiquitous computing has been empowered with the use of temporal series of contexts to organize and analyze the data. This new knowledge research area received the name of Context Histories (*Rosa et al., 2015*; *Martini et al., 2021*; *Aranda et al., 2021*;

*Machado et al., 2021*) or Trails (*Silva et al., 2010*; *Barbosa et al., 2016*; *Barbosa et al., 2018*). This kind of organization allows the exploration of advance strategies to data analysis, such as, profile management (*Wagner, Barbosa & Barbosa, 2014*; *Barbosa et al., 2017*; *Leithardt et al., 2018*; *Dalmina, Barbosa & Vianna, 2019*; *Ferreira et al., 2020*; *Leithardt et al., 2020*), pattern analysis (*Dupont, Barbosa & Alves, 2020*), context prediction (*Da Rosa, Barbosa & Ribeiro, 2016*), and similarity analysis (*Wiedmann et al., 2016*; *Filippetto, Lima & Barbosa, 2021*).

This article presents a model for recommending requirements in software projects, called Nhatos. The proposed model differs from previous literature in exploring the similarity of project context histories to assist RE processes by predicting future contexts. Thus, new requirements are recommended both in the early stages and throughout the project life cycle. The study seeks to answer the following research questions: (1) Is it possible to use project context histories to infer requirements in the requirements identification phase, considering the characteristics and similarity of the projects? (2) Does stakeholder collaboration, providing project characteristics and feedback from recommendations contribute throughout the requirements management processes?

This article has five sections. The next section discusses related works focusing on the scientific contributions. The third section proposes the model, mainly describing its architecture, the similarity analysis strategy, and the proposed Ontology of Requirements Recommendation. The fourth section describes implementation aspects focusing on prototype characteristics, such as technologies, features, screens, and database model. The section focused on evaluation aspects mainly addresses the application of the prototype in two case studies based on 153 real software projects. Finally, the last section presents the conclusion, answers the research questions, and suggests future works.

## RELATED WORKS

The selection of related works demanded the identification of studies that involve the development of models for Requirements Management. The criteria adopted for the choice of works prioritized articles that addressed: (i) models or systems for recommending requirements; (ii) similarity analysis of projects or their requirements; (iii) feedback system on recommendations for new requirements.

*Kim, Dey & Lee (2019)* presented an ontology knowledge base and the design process for recommending security requirements based on the cases of attack and the system domain knowledge. The base has three parts: Ontology APT, Ontology of security general knowledge, and Ontology of domain-specific knowledge. Each ontology can help in understanding the security concerns in their knowledge. Integrating three ontologies with the problem domain ontology allows the derivation of suitable security requirements with the recommendation process of security requirements. The proposed knowledge base and the process can help to derive the security requirements, considering attacks in real systems.

*Liu et al. (2018)* conducted a study that approaches the development or maintaining of Android applications. The authors analyzed the multiple challenges that developers face in creating explanations about permissions use. They proposed a new structure, which

explores possible security requirements recommendations through the description of similar applications. The study uses techniques of information retrieval and text abstract to find frequent uses of permission.

*Xie et al. (2017)* proposed a methodology that uses Conditional Random Fields (CRF) to provide a quantitative exploration of the interactions between users and systems in order to discover potential requirements. By analyzing user behavior patterns at runtime, domain experts made predictions about how users' intentions change. The authors proposed improvements to help address the similar needs identified.

*Bakar et al. (2016)* presented a semi-automated approach, known as Feature Extraction for Reuse of Natural Language Requirements (FENL), for extracting phrases that may represent software resources. The authors aim to extract resources from product reviews online, thus allowing the reuse of software requirements.

*Portugal et al. (2017)* proposed the use of a software versioning repository (GitHub) as a source of information. To deal with large masses of data and provide access to suitable sources, the authors created project profiles with useful attributes for RE. Afterward, they applied clustering and Natural Language Processing (NLP) to recommend projects by identifying similar keywords in their description.

*Williams & Mahmoud (2017)* used the social network *Twitter* as a requirements source to allow a data-driven, interactive and adaptable RE process. The authors performed an analysis with 4,000 tweets from 10 software systems sampled from various application domains. The results revealed that about 50% of the *tweets* collected contained useful technical information. In addition, the results showed that text classifiers like Support Vector Machines and Naïve Bayes can be useful in capturing and categorizing *tweets* technically informative.

*Garcia & Paiva (2016)* presented a recommendation system that collects the history of using a Web service, relates this information to requirements, and generates reports with recommendations that can increase the quality of this service. The proposed approach aims to provide analytical reports in a language close to the business. The system indicates new workflows, navigation paths, identifies potential resources to remove, and correlates the requirements and the proposed changes, helping to keep the specification of the software requirements up to date.

*Hujainah et al. (2021)* proposed a technique for prioritizing requirements and thus selecting the requirements to be developed. While not directly recommending, prioritization helps the selection of requirements and supports the process. The authors addressed this task focusing on specific challenges in this area, such as scalability, lack of automation, and excessive time consumption. The study presented a semiautomated scalable prioritization technique using a multi-criteria decision-making method, clustering algorithms, and a binary search tree. The technique aims to mitigate the need for expert involvement in this process and increase efficiency.

*Swathine & Sumathi (2021)* worked with requirements traceability and based on this information the proposal indicates which requirements must be considered to support the interested parties in the process. This study used a meta-heuristic approach to create a novel traceability system for analyzing systems' functional requirements. The authors

**Table 1  Comparison of related works.**

| Author | Processes | Recommendation | Strategy | Collaboration | Environment | Type |
|---|---|---|---|---|---|---|
| *Kim, Dey & Lee (2019)* | V[1] | R[5] | Ontologies | ✓ | Academic | Experiment |
| *Liu et al. (2018)* | V | R | Description | × | Academic | Experiment |
| *Xie et al. (2017)* | V | W[6] | Historic | ✓ | Academic | Experiment |
| *Bakar et al. (2016)* | E[2] | R | Description | × | Academic | Experiment |
| *Portugal et al. (2017)* | E | P[7] | Commits | × | Industry | Use case |
| *Williams & Mahmoud (2017)* | E | R | Reviews | × | Academic | Use case |
| *Garcia & Paiva (2016)* | E | R | Logs | ✓ | Academic | Experiment |
| *Hujainah et al. (2021)* | V | R | Historic | × | Academic | Experiment |
| *Swathine & Sumathi (2021)* | V | W | Historic | ✓ | Academic | Experiment |
| *Mougouei & Powers (2021)* | E | R | Expert system | × | Academic | Experiment |
| Nhatos model | V, E, S[3], M[4] | R | Context Histories | ✓ | Industry | Use case |

Notes.
  1, Validation; 2, Elicitation; 3, Specification; 4, Management; 5, Requirements; 6, Redefinitions; 7, Projects.

aimed to identify traceable links for supporting decision-making, solving the inconsistency problem, and generating quality requirements.

*Mougouei & Powers (2021)* allowed the selection of requirements considering dependencies and value to be delivered by the requirements. The authors proposed the Dependency-Aware Requirements Selection, an intelligent system that analyzes the value dependencies among requirements, aiming to reduce the risk of value loss. This model considered the user preferences for the requirements, showing promising results in reducing value loss, including when applied in large requirement sets.

Table 1 shows the characteristics adopted in the comparison between Nhatos and related works. The first item (processes) informs which of the RE processes the articles address: validation (V), elicitation (E), specification (S), or management (M). The second item (recommendation) shows the type of item recommended in the study: requirements (R), wrong definitions (W), or projects (P). The third shows the strategic path used by the authors for the recommendations. The fourth item refers to the collaboration of interested parties during the recommendation process. Finally, the last two columns present the environment of the model observation and the type of evaluation.

The analysis of related works indicates four scientific contributions of Nhatos. First, the proposal collaboratively approaches all RE processes, allowing everyone involved to contribute throughout the projects. Nhatos collects different points of view on requirements at any time during the life cycle of the projects, contemplating all RE processes. Second, the model addresses requirements recommendations at the beginning of a new project, using histories of projects already executed, through common characteristics between projects and requirements. Third, the similarity analysis of context histories and NLP allow the recommendation of similar requirements in the initial phase of projects. Finally, Nhatos recommends future contexts based on the similarity analysis of context histories.

## PROPOSED MODEL

According to *Robillard et al. (2014)*, an SERS needs to have specific requirements to be considered a recommendation system, which are: (a) a mechanism for collecting data and artifacts from the development process in a data model; (b) a recommendation mechanism to analyze the data model and generate recommendations; and (c) a user interface to trigger the recommendation cycle and present its results.

Nhatos meets the three requirements mentioned, because: (a) it collects data through a multi-agent system throughout the entire project life cycle; (b) it generates a recommendation considering the current context of the project, and (c) it has an interface on mobile devices to present the results to users and collect their feedback.

In order to measure the applicability of an intelligent tool to support requirements engineers, we conducted a survey involving software design professionals. This research aimed to answer whether the project teams need a proactive tool to support their activities involving the RE processes.

### Principles of Nhatos: survey with 56 professionals

A survey involved 56 professionals working in the software development industry, including project managers, analysts, project teams, and teachers. Participants answered an electronic questionnaire with multiple choice and transcribed questions. About 71% of the interviewees had more than five years of experience in projects. More than 70% of respondents worked in companies with more than 100 employees. The main objective of the research was to capture the perception of professionals regarding the support tools in project management currently used in their work environment. This research allowed to identify gaps and possible improvements in the RE area guiding the specification of Nhatos. The following are the research questions and results:

- *Which areas do you consider most critical to the success of the project?* 50% of respondents selected the scope as the most critical area. Participants also mentioned the areas of Time, Communications, and Integration, with 44.6%, 42.9%, and 39.3% of responses, respectively;
- *In the projects where problems occurred, what were the areas in which the problems were identified?* A total of 39.29% of the participants answered that problems in project management are due to incorrect Scope Management (SM). Other project areas, such as Time Management and Communications, obtained 25 (44.6%) and 21 (37.5%) responses, respectively;
- *What types of suggestions would you like to receive from a proactive project management tool?* According to the interviewees' perception, 32.1% answered that a tool should suggest new requirements for projects;
- *Do you believe that information from other projects already completed could assist in project management?* 85.7% of the members confirmed that history contributes to the management of the new projects.

The perception of the teams collected in the survey allowed to conclude that there is interest from the professionals regarding the use of an intelligent tool to support the

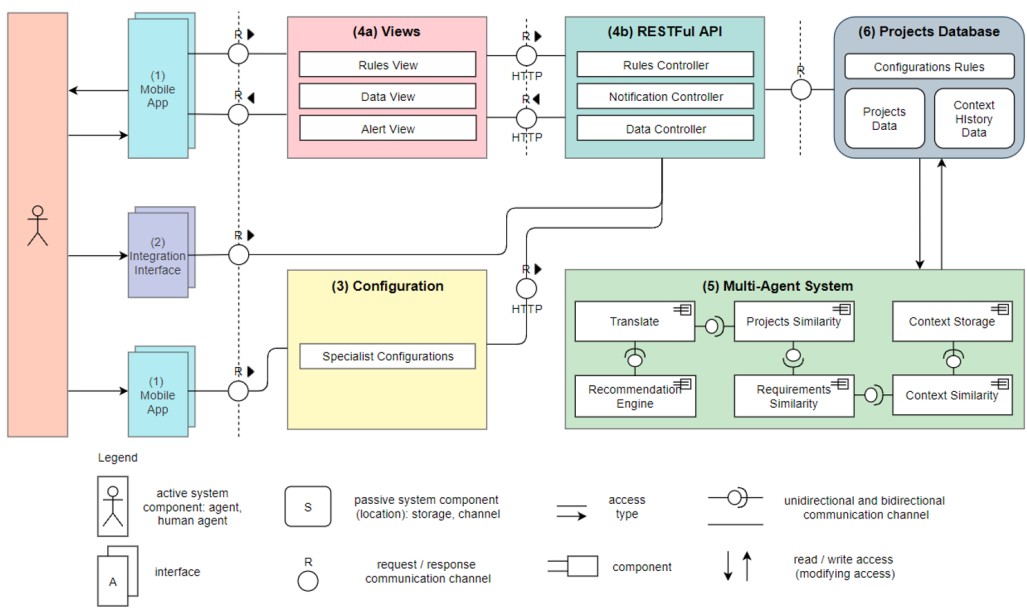

**Figure 1** Nhatos architecture.

project teams during the RE processes. This opportunity stimulated the development of the Nhatos, which aims to assist teams during the RE process life cycle.

## Model architecture

Figure 1 shows Nhatos architecture using the Technical Architecture Module (TAM) modeling specification (*FMC, 2021*). The following components are part of the model and they are seen in the figure with their respective numbering:

1. *Mobile App*: Application in a hybrid structure based on *Javascript* language, which operates both in mobile applications (Android/IOS), as well as in browsers web. Stakeholders use this application during interactions in RE processes. The stakeholders can: (a) add and manage projects and their information; (b) add requirements and modify them; (c) view the recommendations generated by the model; (d) provide feedback about the recommendations received, accepting or rejecting them;

2. *Integration Interface*: It enables the bulk import of data from ongoing projects. When using this system web interface, users may import projects and requirements exported from third-party project management tools, such as *MS Project*, for example;

3. *Configuration*: Project team preferences are entered through an assistant agent, accessed *via* mobile devices. Each project variable receives a weight (area of knowledge, size, methodology, and level of completeness of the schedule). Each recommendation made by the model considers the weight configured by a specialist.

4. *Views* and *API*: These applications operate in an integrated manner in a server environment.

    (a) *Views*: They are characterized by controllers responsible for the business rules of the model, obtaining information already stored in the database to provide information to those involved;

    (b) *API*: A data access interface which uses the *RESTFul* protocol to interconnect the application *Mobile App* and *Database*.

5. *Agents*: Multi-Agent System (MAS) that captures the events related to project's evolution or modification. The capture is triggered when some of these events occur: (a) addition of a new requirement; (b) termination of an activity; or (c) evolution in the percentage of completion of the project. Figure 2 shows the proposed MAS using the the Prometheus methodology (*Larioui, 2020*). The MAS has six agents. The *Translate* agent converts to English the texts from native languages used in the projects. The NLP uses English as the language, so this translation is necessary to Nhatos. *Projects Similarity* analyzes the similarity of the projects using project size, methodology applied and area of expertise. NLP techniques allow to group projects according to their expertise. *Context Storage* stores each event occurrence in the project's history. *Recommendation Engine* permanently monitors the project's events to orchestrate the execution of the other agents when one event occurs. *Requirements Similarity* uses semantic analysis to determine the requirements similarity based on texts written in natural language. This analysis is detailed in the Similarity Analysis subsection. The agent also compares requirements to determine if requirements have the same number of actors. *Context Similarity* performs the similarity analysis in the context histories of the projects.

6. *Projects Database*: It saves application settings, such as (a) Project data; (b) Recommendations made by the model; (c) Feedback from stakeholders regarding the recommendations; and (d) Context histories that occurred throughout the life cycle of the projects.

## Similarity analysis

The similarity analysis occurs in two moments: (1) similarity analysis based on project characteristics, and (2) similarity analysis based on context histories of projects. The first analysis occurs at each insertion of a new project, while the second analysis occurs during a new evolution of the projects' life cycle. Figure 2 shows the representation of the multi-agent system with the three agents that conduct the similarity analysis (Context, Projects and Requirements).

The model captures the characteristics of the new project during its creation. At this time, the project teams have not yet implemented it. The project is in the creation or planning phase. *The Project Management Institute (2017a)* has characteristics that are considered properties of a project. Stakeholders determine these characteristics in the early stages of the life cycle, which are: (a) Area of knowledge; (b) Development methodology (agile, traditional, or hybrid); (c) Level of completeness of the schedule; and (d) Size. Figure 3 shows how the recommendation flow occurs in this step.

After capturing the fundamental characteristics of the project, an expert defines the weights for each one. The specialist defines the importance of each characteristic, based on

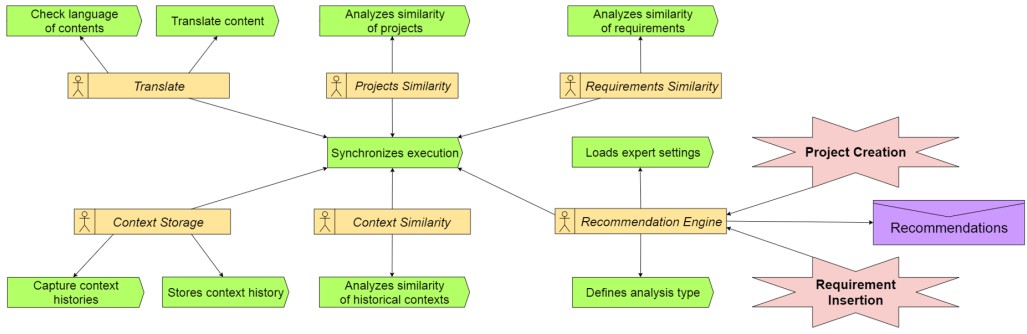

**Figure 2    Multi-agent system.**

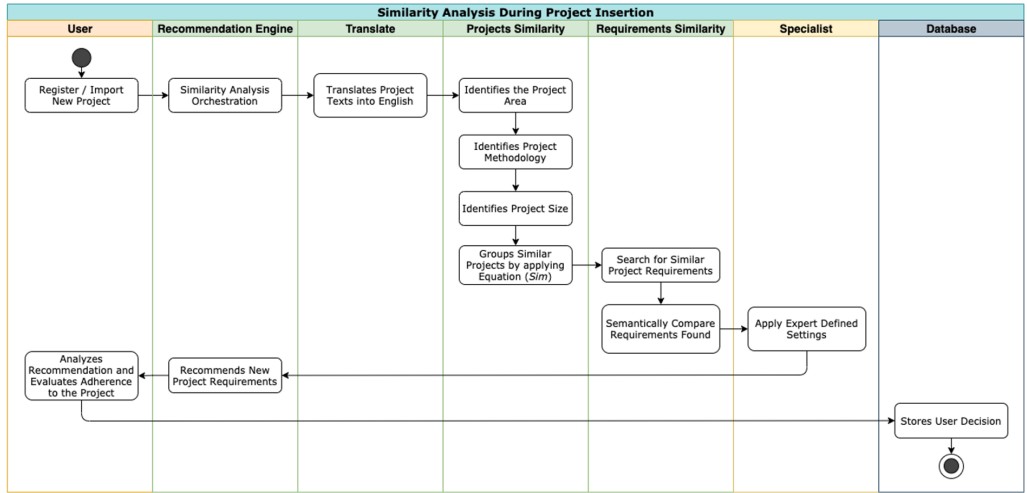

**Figure 3    Similarity analysis by project characteristics.**

its degree of relevance to the project and organization. This definition allows to measure the similarity between the projects before the start of their execution.

After the insertion of a new project, the multi-agent system identifies this event and initiates the recommendation process. The multi-agent system goes through the stored histories and compares the variables with characteristic of each project in the history with the same variables as the original project. The model considers the configurations previously informed by the specialist - these configurations make up a weight system, which will be applied during the calculation.

After the model groups similar projects, the context histories of those projects are analyzed to identify reusable requirements among them. Therefore, for projects in the same group, Nhatos calculates similarity considering the semantic distance between their requirements.

Nhatos defines the semantic distance by analyzing the distance between text documents proposed by *Kusner et al. (2015)*. This approach takes advantage of the results of

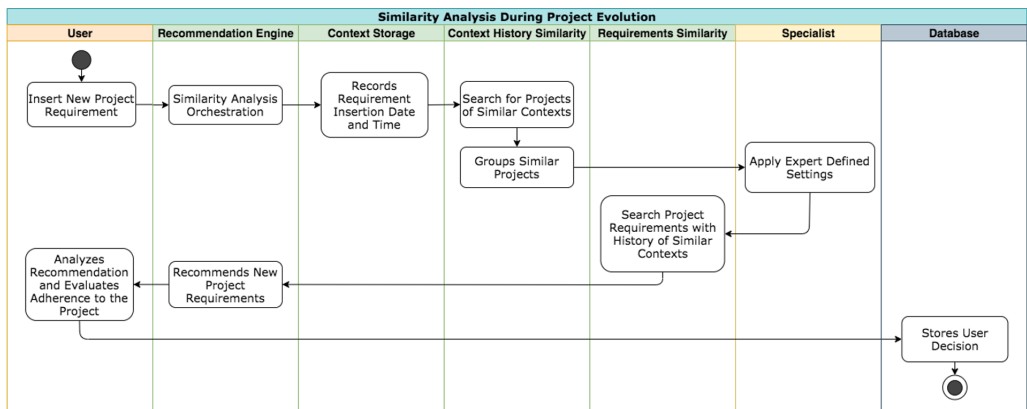

**Figure 4** Similarity analysis by project context histories.

*Mikolov et al. (2013)*, whose model *word2vec* generates combinations of words, on a large-scale ontology, for extensive data sets (for example, we use the training of approximately 100 billion words in this model). In this way, Nhatos compares the texts that describe the objectives of the requirements. Afterward, the results are stored in history, enabling the recommendation of requirements related to the same theme (similar purposes) and similar projects (same area of knowledge).

The recommendation of requirements in the initial phases of the projects aims to bring historical information to the project teams, mainly to the requirements engineers and stakeholders. Then, these users will be able to accept or reject the requirements recommended by the model. Nhatos thus ensures that no requirements of the historical basis are disregarded by those involved during management.

During the life cycle of projects, Nhatos saves the events in context histories. The model identifies information that is susceptible to changes in state over the life cycle of the projects, which are: (a) purpose of the requirements; (b) actors involved in the requirements.

Whenever a user inserts a new requirement into the project or at least one of this context information is modified, the similarity analysis of projects by context histories begins. Nhatos uses the stored histories to complement the similarity analysis based on the characteristics, updating recommendations based on the new information.

Figure 4 shows how the recommendation flow occurs in this step. This flow seeks similar contexts by analyzing the context histories of the project and comparing this information with other stored contexts. Each step presented in the projects' timeline represents information about event occurrences saved in context histories. In this way, this information can later be used by the model to generate a new recommendation. Whenever one of the defined events occurs, Nhatos creates a record in the context history of the project.

Nhatos compares each project context with contexts from similar projects by using the semantic distance between the requirements from the previously-stored histories. The recommendation of the next occurrence of the context history occurs for the project

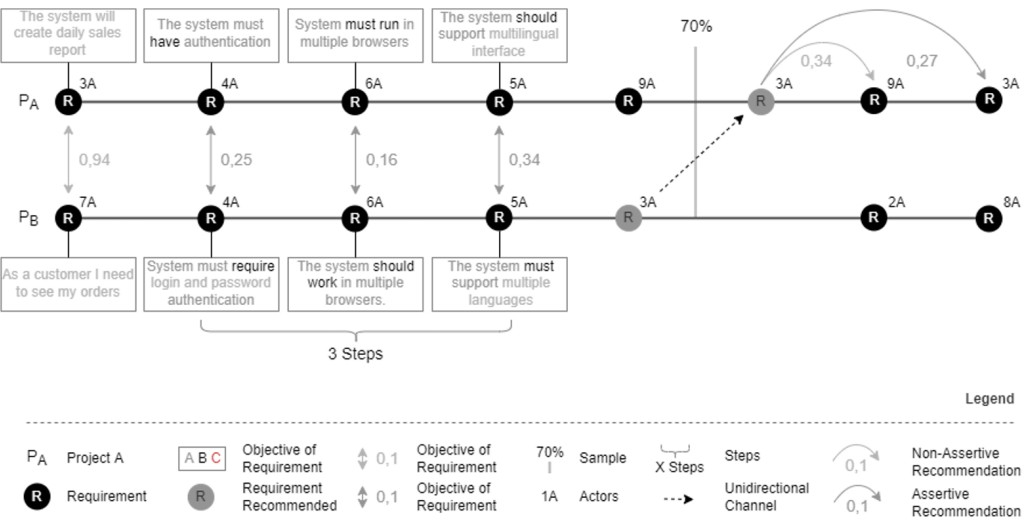

**Figure 5  Similarity analysis by context histories.**

in execution when the distance between the requirements is acceptable (according to the specialist's settings), and the number of actors is equal between the requirements compared.

Figure 5 shows an example of analyzing the context histories of an ongoing project with histories from similar projects. In this example, the entire recommendation flow is elucidated step by step.

We considered two projects to exemplify the similarity analysis of context histories. Both projects (Pa and Pb) have five requirements. The multi-agent system compares requirements between projects in chronological order. In this scenario, the specialist configured the minimum semantic distance of 0.3, training sample 70%, and three similar steps to generate the recommendations. Thus, the comparison of the first requirement of Project A with the first requirement of Project B generated a semantic distance of 0.41 (distance not acceptable according to the configured parameters). Another excluding factor is that these requirements have a different number of actors. Thus, step 1 is not similar. On the other hand, steps 2, 3, and 4 obey both the minimum acceptable semantic distance and the same number of actors involved in the compared requirements. Since the expert has configured recommendations that require at least three similar consecutive steps, Nhatos recommends the fifth requirement of project B for project A.

Assuming that the configured training sample was 70%, Nhatos uses the remaining percentage of the project (30%) to verify that at least one requirement with the same semantic distance and the same number of actors as the recommended requirement occurred throughout the life cycle. Once it occurs, the recommendation made is considered assertive.

The analysis of more than one chronological context, which occurred during the project, aims to identify projects that have a similar execution sequence. This analysis contributes to a higher degree of precision in the recommendations made. The most significant number of similar consecutive contexts indicates the proximity between project implementation.

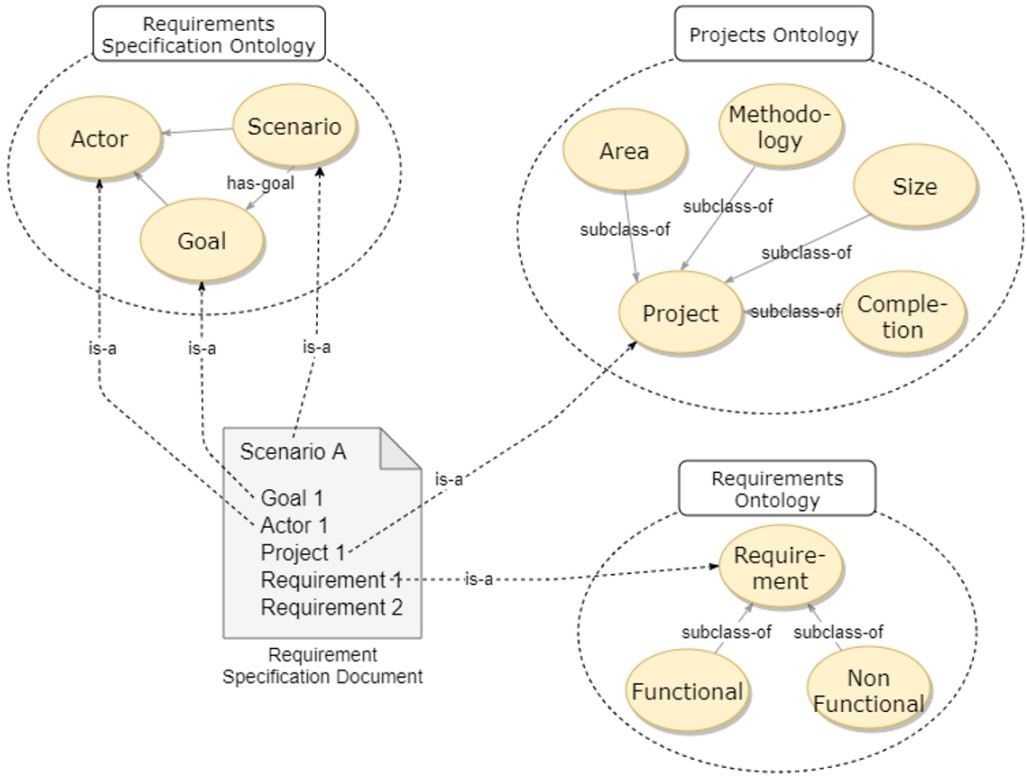

**Figure 6  Ontology of requirements recommendation.**

However, when Nhatos considers more contexts, the fewer projects must be identified as similar and, therefore, the fewer recommendations made, since each project is unique (*Project Management Institute, 2017a*).

## Ontology of requirements recommendation

Figure 6 shows the Ontology of Requirements Recommendation proposed by Nhatos. The domain ontology contains Projects, Requirements, and Specification. This representation is an extension of the work of *Silver (2014)*, with the addition of the *Projects Ontology*. Three ontologies covered the domain considered by Nhatos: (a) *Requirements Specification Ontology* which makes up the requirements specification; (b) *Projects Ontology* which is characterized by the domain of the project and its contextual information; and (c) *Requirements Ontology* that represents the requirements.

The project ontology has the *Project* class composed of the *Area*, *Completion Level*, *Methodology* and *Size*. These subclasses define the contextual information of the projects. The ontology that comprises the requirements specification has the classes *Scenario*, *Goal*, and *Actor*. These classes represent the elicitation process, identifying new requirements and relating them to projects. The ontology that determines the requirements has the class *Requirement* and its subclasses *Functional Requirement* and *Non Functional Requirement*, which define the types of requirements.

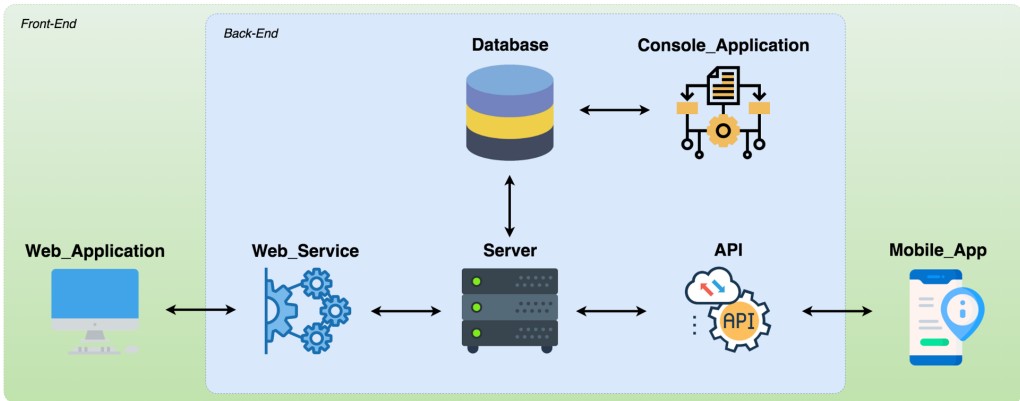

**Figure 7  Prototype overview.**

We added a specification document (*Requirement Specification Document*) to exemplify how the requirement is instantiated when using ontologies. Each document has a scenario, where the actors through the defined objectives identify requirements for a given project.

## IMPLEMENTATION ASPECTS

We developed a software prototype to meet the model definitions. Figure 7 shows that the prototype integrates three applications: (1) *Console Application* (https://github.com/robsonklima/nathos-py-v2); (2) *RESTful API Application* or *WebService* (https://github.com/robsonklima/nhatos_api); and (3) *Hybrid Application* (https://github.com/robsonklima/nhatos_front_end), composed by the Web Application and the Mobile App. The first two are back-end applications, which run on a server. The third application operates on mobile devices, acting as front-end software. The users involved in the requirements engineering processes used this application.

### Hybrid application

The *Hybrid Application* runs both on mobile devices and on conventional computers operating in a browser web. In this way, the application has two interfaces for communication with system users: (1) *Web Application*; and (2) *Mobile App*.

This application allows the interaction of project teams with Nhatos, allowing the management of projects, requirements, activities, resources, registration of interested parties, and evaluation of recommendations. The prototype allows to monitor projects, capturing context information to compose the context histories. The software can be used throughout the life cycle of projects. Further, this application presents the recommendations to the user, thus allowing the collection of feedback from interested parties.

Figure 8A shows the interface for the project presentation with characteristic information about the project such as size, methodology, percentage of evolution, and area of knowledge.

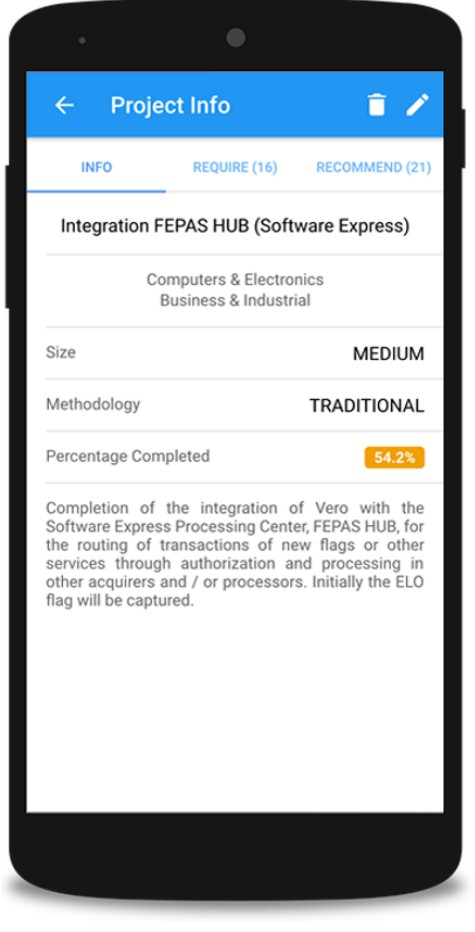
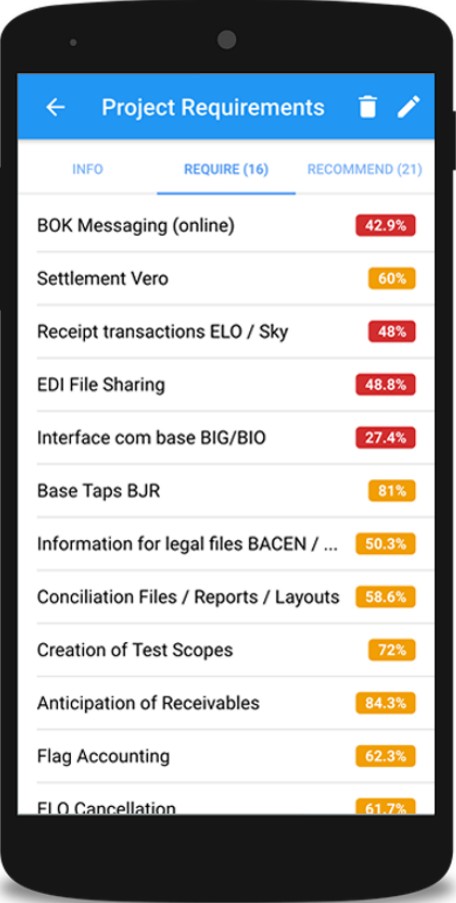

(a) Project and Information                    (b) Requirements and Percentage of Activities

**Figure 8** Screenshots with project details and requirements.

Figure 8B presents the list of project requirements, as well as their respective percentage of evolution.

Figure 9A shows the settings of the weight variables, which are defined by an expert. These variables define the importance of each aspect of the project and its requirements during the recommendation process. The user can define weights related to project area, size, methodology, and level of completion, as well as the acceptable semantic proximity between the requirements that will be considered for a possible recommendation.

Figure 9B presents examples of recommendations. The interface contains the recommended requirement, as well as the requirement that raised this recommendation and the semantic distance between these requirements. The interaction area also enables users to provide their feedback, selected from the options to accept or reject the recommendation. This user decision is registered in order to evaluate the acceptability of the recommendations by stakeholders in the future.

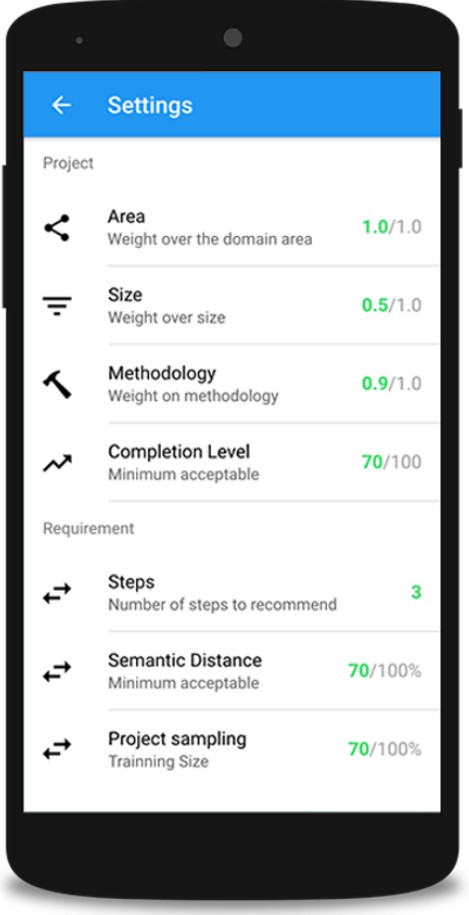 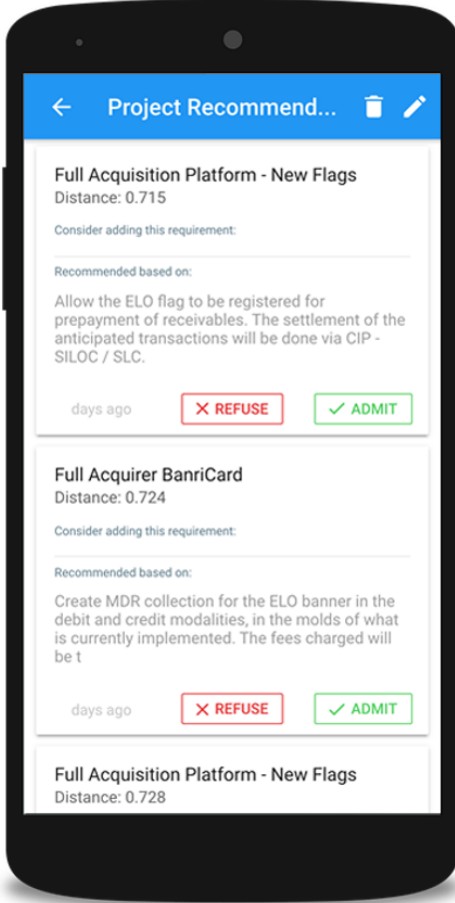

(a) Configurations      (b) Evaluation of Recommendations

**Figure 9**    Screenshots with specialist settings.

### Console application

This application developed in *Python* is an encapsulated software that acts in the form of service. The software uses the concept of multi-agent systems, proposed by *Padgham & Winikoff (2004)* to implement the *Agents* layer. The application does not depend on direct interaction with the users, being triggered by modifications detected in the environment. The *Agents* layer implements the following 6 agents: (1) *Recommendation Engine*; (2) *Translate*;(3) *Projects Similarity*; (4) *Requirements Similarity*;(5) *Context Similarity*; and (6) *Context Storage*.

The *Recommendation Engine* runs systematically through the detection of changes occurred in the environment. The agent always activates when a user inserts a new project or requirement. Once performed, this agent is responsible for starting the execution of all other agents.

The *Translate* agent translates all the contents entered by the user into the English language, enabling the execution of subsequent agents. Since, as a premise, all contents

must be registered in English. Nhatos uses NLP and the *corpus* of texts obtained for the application of the study is written in this language. The software loads the projects and their requirements from the database, translating them using the *API Google Cloud Translate* (*Google, 2021b*). We considered the usage of this API does not generate discrepancies in translations since this API has an estimated accuracy of 85% (*Aiken, 2019*) and the texts translated are technical, having a technical writing pattern.

The *Projects Similarity* agent activates after the translation of the content of the projects and their requirements. This agent analyzes the similarity between all projects in the database and groups the characteristics of the projects separately. The software considers the information in the ontology (Fig. 6), considering all projects in the database according to size, area of knowledge, management methodology, and level of completeness (schedule). After consulting all projects, the agent classifies and labels each project, so that the next step of the algorithm starts.

The *Requirements Similarity* employs the use of NLP to find requirements that contain equivalent objectives, as well as the same number of actors involved. This agent also considers the similarity analysis between the projects, performed previously. In this way, the algorithm analyzes the similarity between the requirements of projects considered similar.

First, the agent appropriates the new settings for distance, steps, and sampling, thus starting new processing. Then, it removes the previous recommendations (if any), to start a new recommendation process. This method removes all recommendations that originated from the same distance, number of steps, and sampling setting. Because, it is considered that throughout the life cycle, the requirements may have changed regarding their objectives or actors involved (*Project Management Institute, 2017a*).

Then, the agent retrieves all projects from the database. The requirements of each project are obtained. Each requirement is compared with the requirements of similar projects. In this step, the agent checks whether the objective of both requirements meets the established distance parameters, as well as the number of actors. Finally, once the number of requirements found sequentially is equal to the number of configured steps, this requirement is considered recommendable to the project.

The *Context Similarity* agent analyzes whether the recommendations carried out within the sample selected by the specialist occurred in the project. The software loads all recommendations made and it selects the project requirements after the recommendation's record date. If among the requirements, a requirement is found with the same semantic distance to the compared requirement and, containing the same number of actors, this recommendation is considered assertive. Otherwise, the recommendation becomes non-assertive. Finally, this decision is stored in the database.

The *Context Storage* agent keeps the information in a database with four main entities: (1) *projects*; (2) *requirements*; (3) *requirements distance*; and (4) *recommendations*.

Figure 10 shows the relational entity diagram of the model database. The *projects* entity saves information related to projects. This entity registers basic project information and

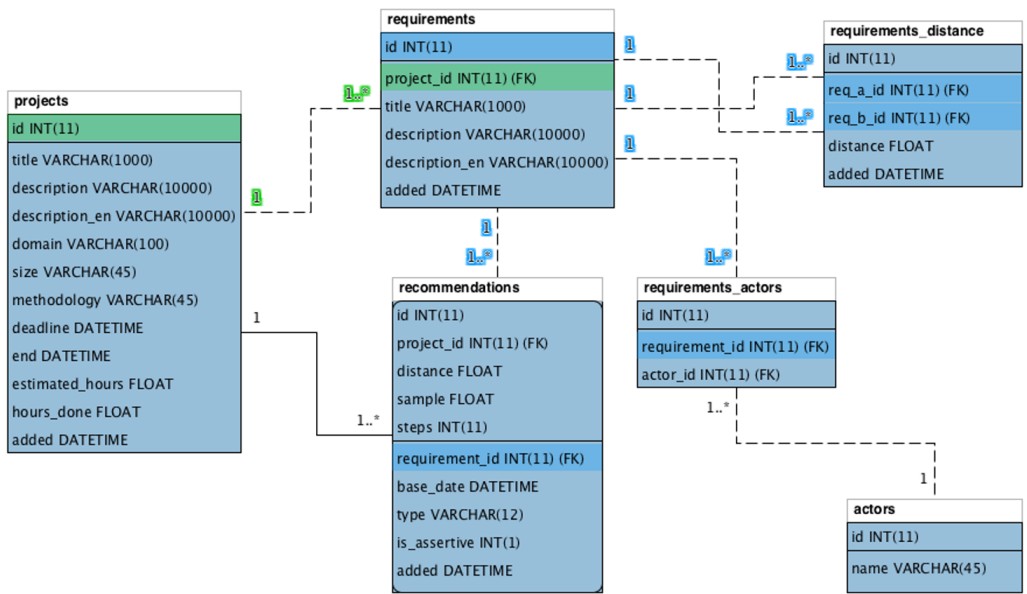

**Figure 10   Relational entity of the Nhatos model.**

its characteristic information, such as: opening statement (*description*), knowledge area (*domain*), size (*size*), and methodology applied during its development (*methodology*).

The *requirements* entity keeps model requirements. The same project can contain several requirements, according to the relationship in the diagram (1..n). The actors of each requirement are stored in the entity *actors* and an actor can be linked to several requirements and vice versa, as shown in the entity *requirements_actors* (n..n).

The *requirements_distance* entity saves the distance information between the processed requirements. The entity stores the original requirement (*req_a_id*) and the compared requirement (*req_b_id*), as well as the respective semantic distance (*distance*) between them.

Finally, the *recommendations* entity keeps the recommendations inferred by Nhatos. Each recommendation is directed to a project (*project_id*) and the requirement that generated such a recommendation (*requirement_id*). The assessment of the assertiveness of each recommendation is stored in the *is_assertive* property.

### RESTFull API application or web service

The *RESTFull API Application* provides a communication channel between the *Hybrid Application* and the *Context Storage* application. Using the *RESTFull* protocol, the application allows data traffic in the *JSON* format between applications. It enables the exchange of data between the hybrid application, used by users, and the information already processed by *Console Application*, which stores its information in the Nhatos model database.

**Table 2  Profile of participating teams.**

| Team | Role | Experience (Years) | Mode |
|---|---|---|---|
| Team A | Scrum master | 15+ | Local |
| | Product owner | 10+ | Distributed |
| | Designer | 5+ | Local |
| | Developer | 10+ | Local |
| | Developer | 5+ | Local |
| | Developer | 5+ | Local |
| | Test analyst | 5+ | Local |
| Team B | Project manager | 25+ | Distributed |
| | Developer | 10+ | Distributed |
| | Developer | 5+ | Local |
| | Developer | 5+ | Local |
| | Test analyst | 5+ | Local |

# EVALUATION ASPECTS

The application of a case study in a software development company allows answering the research questions. This company develops application solutions for a banking institution. This study aimed to confirm the hypothesis of using the analysis of context histories of projects to recommend requirements for new or ongoing projects.

The study employed a database with the context histories of 153 software development projects. The database has projects with different resources and development methods, such as distributed or local teams. The use of different characteristics of the 153 projects allowed the analysis of a diversity of contexts to the recommendation of requirements.

The evaluation considered two cases: (1) two teams evaluated the use of the prototype during the implementation of 5 real projects, and (2) 17 completed projects were used to evaluate the recommendations made by Nhatos, comparing the recommendations done with the requirements in the 17 original projects. Next subsections describe these evaluations.

## Team evaluation during project execution

The first case involved two teams with a total of 12 professionals. These professionals were asked to validate the recommendations made by the model. Table 2 shows the profile of the teams that participated in this experiment.

Initially, project teams inserted information related to the projects in the database (scope and descriptive information, terms of reference, resources, schedule, and tasks/activities). The professionals used the integration interface to insert information into the database. After, the project teams identified and registered the requirements using the prototype running on mobile devices.

The similarity analysis of the projects used NLP since approximately 79% of the requirements and project documents are written in natural language (_Luisa, Mariangela & Pierluigi, 2004_). The algorithm classified each project through the use of NLP according to its respective area of knowledge. The identification of these areas involved the use of the

**Table 3  Recommendations made by project.**

| Project | Area | Recommendations | Accepted | % Accepted |
|---|---|---|---|---|
| Renegotiated operations - Restructured | Finance | 16 | 11 | 68.7 |
| Alteração renov autom cheque especial PF | Finance | 14 | 8 | 57.1 |
| CDB movements in M-BANKING | Business & Industrial | 9 | 7 | 77.7 |
| Parameterization of indexers | Finance | 10 | 9 | 90.0 |
| Automatic lock renewal | Finance | 13 | 8 | 61.6 |
| Approval percent | | | | 71.0 |

opening statement. In addition, the project charter contains a high-level description of the project. The classification used the *Google Natural Language* API (GNL), which provides resources for the analysis of unstructured texts, such as content classification and entity identification. The content rating analyzes a document and results in a list of categories that apply to the found text. The classification can still contain several levels, specifying the greater depth of details about the area of knowledge in question (*Google, 2021a*).

Currently, GNL processes English sentences only. The projects had information in Portuguese, so it was necessary to previously translate the descriptive content of the projects before carrying out the classification process. The prototype performed the translation automatically using the *Google Cloud Translation* API (GCT) (*Google, 2021b*). The GCT receives a phrase as input, identifies its language and translates it into the language selected by the user. During the study, the agent translated all sentences into English. After translation, Nhatos categorized the projects using GNL based on the 153 projects. This step classified 23 projects as Finance (15.03%), 20 as Business and Industry (13.07%), 10 as Computers and Electronics (6.53%), 6 as Credit and Lending (3.92%), and 4 as Accounting and Auditing (2.61%), having these categories the highest number of classified projects.

In addition, the variables registered in the *Configuration* module allowed to defining the knowledge area of each project. The similarity analysis used this definition. These variables received weights, which reflect the uniqueness of each project. The weight setting allows the algorithm to generate different recommendations according to the characteristics recorded by the specialist. Table 3 presents the recommended requirement values for each project and the requirement values added to new projects through the recommendations.

During the case study, users included in the projects twenty requirements, which were used to evaluate the recommendations made in the execution of the project. Whenever a user added a new requirement, the *Requirements Similarity* Agent identifies the event and performs the semantic proximity analysis. The Agent compares the new requirement description with the requirements stored in the context history of the project.

The GNL algorithm analyzes texts in English, performing a semantic analysis. Thus, the model translates the description of the project or requirements which are in another language. Table 4 shows an example of a requirement added to a project and the recommended requirements based on this insertion. Nhatos considers the objective of each requirement and also the number of related actors.

**Table 4  Semantic analysis for requirements recommendation.**

| Included requirement | Recommended requirements | Distance |
|---|---|---|
| The software must allow the manager to request airline tickets (1 actor). | The software must allow the manager to order supplies (1 actor). | 0.21 |
| | The software must allow the manager to request resource transfers between projects (1 actor). | 0.32 |

At this stage, Nhatos applies the analysis to the entire database, regardless of whether projects were compared in the first stage of the recommendation. The semantics of the included text and the objectives of the requirements may change throughout the project (*Dick, Hull & Jackson, 2017*). This step also allows project requirements that were not originally recommended to be analyzed and considered. The analysis is carried out at this point on the requirements, considering the grouping of projects based on their characteristics. The semantic distance represents the comparison of the requirement objectives, having a floating-point value between 0.0 and 1.0. The distance closer to 0 indicates that the recommended requirement is semantically closer to the original.

Consequently, the closer the semantic distance to 1, the less similar the recommended requirement is considered when compared to the original. The actors of each requirement are also considered during this stage of the analysis. This context information considers the number of actors to which a requirement is related.

During the case study, for each of the five projects, the model analyzed the similarity, recommending the requirements between similar projects. Soon after, the team analyzed the recommended requirements for the five registered projects. Table 3 shows that the approval rate of the requirements had an average of 71.0%. The average appropriation of the recommendations presented for new projects shows the acceptance of the requirement recommendation model evaluated by the teams, in order to provide more information to the managers since the beginning of the project.

## Evaluation of recommendations through analysis of context histories

The second case compared the requirements registered in 147 completed projects with the requirements recommendations made by Nhatos. This study allowed to infer whether the recommendations made, considering a sample of 70% of the progress of the projects, were in fact inserted into the remaining percentage of the project.

This scenario evaluated the recommendations made in many situations considering projects with different characteristics. Most of the projects (81) used the agile methodology based on the framework SCRUM (*Sutherland & Coplien, 2019*). The others (66) employed traditional methodology based on the good practices proposed by *Project Management Institute (2017a)*. We classified the projects into one of three size categories, considering the execution time of each one: small (up to 500 h), medium (up to 3,000 h), and large (over 3,000 h).

**Table 5  Evaluation of recommendations made by the Nhatos.**

| # | Distance | Steps | Sample | Non-assertive recommendations | Non-assertive recommendations (%) | Assertive recommendations | Assertive recommendations (%) | Total |
|---|----------|-------|--------|------------------------------|-----------------------------------|---------------------------|-------------------------------|-------|
| 1 | 0,25 | 3 | 0,7 | 248 | 51,56 | 233 | 48,44 | 481 |
| 2 | 0,25 | 4 | 0,7 | 19 | 70,37 | 8 | 29,63 | 27 |
| 3 | 0,25 | 5 | 0,7 | 1 | 50 | 1 | 50 | 2 |
| 4 | 0,3 | 3 | 0,7 | 555 | 29,65 | 1317 | 70,35 | 1872 |
| 5 | 0,3 | 4 | 0,7 | 194 | 30,03 | 452 | 69,97 | 646 |
| 6 | 0,3 | 5 | 0,7 | 56 | 28,43 | 141 | 71,57 | 197 |
| 7 | 0,35 | 3 | 0,7 | 904 | 16,96 | 4427 | 83,04 | 5331 |
| 8 | 0,35 | 4 | 0,7 | 443 | 17,51 | 2087 | 82,49 | 2530 |
| 9 | 0,35 | 5 | 0,7 | 237 | 17,52 | 1116 | 82,48 | 1353 |

The learning phase used a sample of 70% referring to the execution of each project. The Nhatos Learning consists of collecting the project's evolutionary events. Figure 4 shows this process in the Recommendation Engine, Context Storage, and Context History Similarity steps. With this learning, Nhatos generated requirement recommendations for the same projects. The remaining 30% of data from the executed process was the base for assessing the recommendations made by the model.

The similarity analysis between the projects taking into account the context histories considers that a consecutive sequence of contexts must be similar. The model then recommends the next requirements for the project being implemented based on the sequence of events that generated context histories.

In all, nine test scenarios were configured using different parameterizations to find the best scenario of recommendations, where each test considered the entire historical database. Table 5 shows the results of the different settings applied. A test scenario addresses different combinations of the variables *Distance*, *Steps* and *Sample*.

All test scenarios considered a 0.7 sample value. Therefore, the training sample to generate the recommendations contained 70% of the evolution of the projects' schedule (life cycle). Tests 1, 2 and 3 considered configurations with a minimum semantic distance between the requirements of 0.25 (75%). All 3 step configurations for these tests resulted in an assertiveness rate equal to or less than 50%. Scenario 2 and 3 generated a few recommendations, 27 and 2, respectively. Test 1 generated a total of 481 recommendations. However, as mentioned, the assertiveness did not reach 50%.

On the other hand, tests 7, 8, and 9 generated a significant amount of recommendations. The assertiveness rate was high, above 80% in the three cases. However, the semantic distance proved to be relatively comprehensive, considering requirements only 65% similar. Therefore, the tests allowed the inference of a large number of requirements recommendations (a total of 9,214). In these scenarios, Nhatos provided a high number of recommendations, being more than the teams could assess. Therefore, the model did not attend to a requirement of ubiquitous applications in these cases since these applications must be minimally intrusive (*Satyanarayanan, 2001*).

Scenarios 4, 5, and 6 were more promising than the first ones. All three achieved a percentage rate of assertiveness close to or higher than 70%. However, only scenarios 4 and 5 had an adequate number of recommendations for analysis in the use case.

The configuration of the Scenario 4 considered a minimum distance of 0.3 (70%) and three steps, obtaining a hit rate of 70.35% for assertive recommendations. Of the 1,872 inferred recommendations, 1,317 were correct, while 555 were incorrect. Scenario 5 considered a configuration of a minimum distance of 0.3 (70%) and four steps. This scenario obtained a hit rate of 69.97% of assertive recommendations. A total of 452 of the 646 recommendations were correct, while 194 were unsuccessful. The different scenarios allowed a new round of tests to be carried out, obtaining the most certain configuration for the database.

## CONCLUSION

This article proposed Nhatos, a computational model which provides recommendations considering the characteristics of each new project. In this way, the teams start the life cycle of each project with a broader set of information, making the project planning more assertive, which increases the chances of success.

In addition, Nhatos infers new recommendations during the execution of the projects through the analysis of the included requirements. The recommendations benefited from a semantic analysis of the text that understood the requirements' objectives, as well as the number of actors involved. In this sense, new scenarios for the projects are considered during their implementation. The model considers context histories of projects when recommending new requirements, when considering the schedule of similar contexts and when compared to the original projects.

The research questions allow to validate the use of the Nhatos in two dimensions: (i) requirement recommendation considering the context histories of the projects; (ii) elicitation and specification of requirements, allowing their use collaboratively. In this sense, the results demonstrated the adherence of the Nhatos to proactive requirements management in projects. A summary of the main conclusions is as follows:

1. Nhatos achieved an accuracy of 65.33% regarding the average value of the 9 test scenarios performed and scenario 7 reached an accuracy of 83.04%.

2. The first research question focused on the suitability of the recommendations made by the model for the new project, considering the team that developed the projects and the projects already executed. The evaluation confirmed the relevance of projects' context histories for recommending requirements for the projects. Case 1 presented an average recommendation approval rate of 71.0%, proving that Nhatos can make suitable recommendations based on experts' configurations and characteristics of other projects. Case 2 proved as true the hypothesis of using the context histories for the requirements recommendation, achieving a value higher than 80% of assertiveness in different scenarios through the similarity analysis of the project context histories.

3. The second research question assessed the ability of the model in recommending requirements throughout the projects' life cycle in a collaborative manner. During the

project's follow-up period in the case study, the registration of 20 new requirements occurred, in addition to the requirements recommended by Nhatos. All team members participated in a collaborative analysis of each requirement and contributed knowledge during the elicitation, specification, and validation processes, providing more information for the project. The collaboration of the requirements management team allowed the evaluation of possible impacts that may occur in the projects. Also, teams could collaborate during all requirements management processes through the use of the prototype.

4. The answers to the research questions confirm the main scientific contributions of this study, which is the recommendation of requirements considering the characteristics of the projects and analyzing the context histories, in addition to monitoring the entire life cycle of the requirements throughout the project. Thus, the model helps in planning projects by providing a broader set of information adhering to the project in progress for requirements engineers when starting a new project.

5. The collaboration of all interested parties enhanced the model, mainly in the identification and specification of requirements. This is not present in the related works. This differential enabled the collection of more information during the implementation of the project and brought technical and practical knowledge about the importance of the requirements by all stakeholders.

6. The case studies and the prototype allowed the evaluation of the Nhatos, contributing to the observations of gaps in the management of project requirements. The case studies focused on answer the two research questions presented in the introduction.

Based on the results obtained in the case studies, we suggest the following opportunities that future studies can explore:

1. The exploration of the model usage, considering projects from different companies.

2. The monitoring of the prototype use over time since the prototype can provide a more robust design history and more assertive requirement recommendations, considering the model benefits from a growing database.

3. The exploration of the pattern analysis of context histories, which can allow detecting emergent patterns related to requirements and projects.

4. Future investigations can enhance the interface among applications, performing a deeper analysis concerning the API built.

5. Future studies may perform a wider comparative analysis with related works to investigate the results obtained in this study, exploring the data used and the saved time through the use of Nhatos.

## ACKNOWLEDGEMENTS

We would like to thank University of Vale do Rio dos Sinos (UNISINOS), VALORIZA, and COPELABS for embracing this research.

### Funding

This work was supported by national funds through the Fundação para a Ciência e a Tecnologia, I.P. (Portuguese Foundation for Science and Technology) by the project UIDB/05064/2020 (VALORIZA—Research Centre for Endogenous Resource Valorization), and Project UIDB/04111/2020, ILIND–Instituto Lusófono de Investigação e Desenvolvimento, under project COFAC/ILIND/COPELABS/3/2020. There was no additional external funding received for this study. The funders had no role in study design, data collection and analysis, decision to publish, or preparation of the manuscript.

### Grant Disclosures

The following grant information was disclosed by the authors:
National funds through the Fundação para a Ciência e a Tecnologia, I.P. (Portuguese Foundation for Science and Technology): UIDB/05064/2020.
VALORIZA—Research Centre for Endogenous Resource Valorization: UIDB/04111/2020.
ILIND–Instituto Lusófono de Investigação e Desenvolvimento: COFAC/ILIND/-COPELABS/3/2020.

### Competing Interests

The authors declare there are no competing interests.

### Author Contributions

- Robson Lima and Alexsandro S. Filippetto conceived and designed the experiments, performed the experiments, analyzed the data, performed the computation work, prepared figures and/or tables, authored or reviewed drafts of the paper, and approved the final draft.
- Wesllei Heckler analyzed the data, prepared figures and/or tables, authored or reviewed drafts of the paper, and approved the final draft.
- Jorge L.V. Barbosa conceived and designed the experiments, analyzed the data, performed the computation work, authored or reviewed drafts of the paper, and approved the final draft.
- Valderi R.Q. Leithardt analyzed the data, authored or reviewed drafts of the paper, and approved the final draft.

### Data Availability

The raw data is available in the Supplemental File.

The Console Application is available at GitHub: https://github.com/robsonklima/nathos-py-v2

The RESTFul API Application or WebService is available at GitHub: https://github.com/robsonklima/nhatos_api

The Hybrid Application is available at GitHub: https://github.com/robsonklima/nhatos_front_end

## Supplemental Information

Supplemental information for this article can be found online at http://dx.doi.org/10.7717/peerj-cs.794#supplemental-information.

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
