# Peer review of "Towards ubiquitous requirements engineering through recommendations based on context histories"

_PeerJ Computer Science, doi:10.7717/peerj-cs.794_

## Round 0.1 · original submission · Minor Revisions

We can accept this paper if you carefully address all the comments provided by the reviewers.

Reviewer 1 ·

Basic reporting

1. the work is prepared well and having good background of the issues considered in the proposed model
2. Nhatos architecture using the Technical Architecture Module (TAM) modeling specification has been considered, which is right move

Experimental design

1. the experiments made in the work are on the following nodes:
a. Relational Entity of the Nhatos Model
b. Profile of Participating Teams
c. Recommendations Made by Project
d. Recommendations Through Analysis of Context Histories
2. The RESTFull API Application provides a communication channel between the Hybrid Application and 421 the Context Storage application has been implemented using right approach, however, more consistency in analysis is needed
3. Learning phase steps to be explained in better way

Validity of the findings

findings have been validated and aps have been consistent
the data used and provided is consistent, however, more comparative study is needed

Additional comments

need to have relook at comparative study

Reviewer 2 ·

Basic reporting

# Literature work should be improved in the paper as it is very limited. Add one or Two table of comparison along with line diagram of existing work.

# The proposed methodology and design must be compared with existing work. Dedicated separate sub-section for this. Also, use block/line diagram for such proposed methodology.

Experimental design

# The work is under the Aims and scope of this journal and also, presented with scientific contribution of this study as use of the similarity analysis of this work.

# The article proposed "Nhatos" which is a computational model for ubiquitous requirements management . This is very appreciated.

# The experiment demonstrated that the model achieved more than 70% stakeholder acceptance of the recommendations. This is justified

Validity of the findings

# Conclusion of this paper can be written in systematic way

Reviewer 3 ·

Basic reporting

The paper titled, "Towards ubiquitous requirements engineering through recommendations based on context histories" proposes a technique to recommend requirements on the basis of context histories of projects. The paper is generally well written and covers literature in an adequate manner. The authors also provide a comparison of related works with the proposed approach and the developed tool in tabular form (Table 1) which shows the usefulness of the Nhatos model as compared to other similar methods in the literature.

The figures in the paper are generally in good shape however, Figures 1 and 3 give a blurred look when the pdf of paper is zoomed in for better reading perhaps their quality can be improved.

The raw data has been shared and the interpretation of the approach and results are self contained. However, I could not open the links (https://github.com/robsonklima/nhatos api and
https://github.com/robsonklima/nhatos front end) which is a requirement for this journal that a working software version should be available in an online repository.

Experimental design

The research proposed in this paper is definitely within the scope of the journal and the research questions identified are well defined, relevant and meaningful filling the gap of recommending requirements on the basis of context histories of projects. However, I have the following questions for the authors:

1) The basic project information was in Portuguese and it was translated to English using the Google Cloud Translation API. The accuracy of this API is 100%, how did the authors make up for any discrepancies in the translation and how potentially this would have impacted the results currently produced by Nhatos?

2) On page 6 of the paper at line 234, the authors talk about the semantic similarity of requirements during the analysis of requirements. However, it will be great if they can shed some light on which semantic similarity measures for NLP have been used and why plus whether there was any syntactic analysis done prior to it or not?

Validity of the findings

Since, I was unable to find and download from these two links (https://github.com/robsonklima/nhatos API and https://github.com/robsonklima/nhatos front end), therefore, it is difficult to comment on the replication and reproducibility of results. In the next version I would expect that the link is live and the files retrievable from these links.
On page 14, Table 3, the data provide show acceptance percentages of five different projects none of which is above 68.7% the last row of the table shows the approval percent which is computed to be 75.1%. How is this approval percent computed and how does it go well above the accepted requirements percentage?
Second last row of Table 3, shows an acceptance percent value of 61,6 which should be 61.6
The conclusions are well stated and linked to research questions along with providing possible future directions. However, my final question authors is how much would we save in terms of time by using Nhatos model specifically given the fact that it some times generates recommendations too many to be assessed by the team members as mentioned on page 15, lines 529-30?

---

## Round 0.2 · accepted · Accept

Congratulations, the manuscript is now acceptable for publication.